# Infrared laser-induced gene expression for tracking development and function of single *C. elegans* embryonic neurons

Anupriya Singhal[1] & Shai Shaham[1]

Visualizing neural-circuit assembly *in vivo* requires tracking growth of optically resolvable neurites. The *Caenorhabditis elegans* embryonic nervous system, comprising 222 neurons and 56 glia, is attractive for comprehensive studies of development; however, embryonic reporters are broadly expressed, making single-neurite tracking/manipulation challenging. We present a method, using an infrared laser, for reproducible heat-dependent gene expression in small sublineages (one to four cells) without radiation damage. We go beyond proof-of-principle, and use our system to label and track single neurons during early nervous-system assembly. We uncover a retrograde extension mechanism for axon growth, and reveal the aetiology of axon-guidance defects in *sax-3*/Robo and *vab-1*/EphR mutants. We also perform cell-specific rescues, determining DAF-6/patched-related site of action during sensory-organ development. Simultaneous ablation and labelling of cells using our system reveals roles for glia in dendrite extension. Our method can be applied to other optically/IR-transparent organisms, and opens the door to high-resolution systematic analyses of *C. elegans* morphogenesis.

[1] Laboratory of Developmental Genetics, The Rockefeller University, 1230 York Avenue, New York, New York 10065, USA. Correspondence and requests for materials should be addressed to S.S. (email: shaham@rockefeller.edu).

The assembly of neural circuits requires precise and orderly axon navigation, resulting in apposition of neurons that will become synaptic partners. While guidance molecules directing neurodevelopment have been extensively explored[1,2], real-time tracking of single neurites during nervous system assembly has been limited by the size and morphological complexity of nervous systems, as well as by technical challenges preventing high-resolution observation of neurite growth in living specimens.

The embryonic *Caenorhabditis elegans* nervous system contains only 56 glia and 222 neurons, many of which form synaptic contacts in a brain neuropil called the nerve ring (NR). This small cell complement makes *C. elegans* attractive for comprehensive studies of nervous-system assembly. Onset and termination of neurite navigation in *C. elegans* are delimited by the spatially/temporally invariant cell lineage[3,4], which generates neurons of characteristic morphologies; and a synaptic connectome, characterized at high resolution by electron microscopy[5]. However, events bridging outgrowth initiation and circuit incorporation are known only for a few neurons. Technical challenges hamper comprehensive analyses of these critical intermediate steps. Embryos are photosensitive at GFP excitation wavelength[6], making long-term high-resolution imaging tricky; and embryos move rapidly, making volumetric imaging susceptible to motion-induced blurring. New imaging technologies, such as light-sheet fluorescence microscopy, and faster cameras address some of these challenges[7,8].

Nervous-system anatomy also provides a major roadblock for imaging neuron development. Neurites fasciculate in commissures, and since each neurite has a diameter smaller than the diffraction limit, tracking neurites optically is impossible if more than one per fascicle is labelled. This problem was already appreciated by Cajal, who used the inefficiency of Golgi staining to systematically catalogue mammalian and insect nervous systems[9]. Thus, sparse labelling is essential for a developmental catalogue of nervous-system assembly. Yet, most embryonic reporters in *C. elegans* and other animals are broadly expressed. Stochastic combinatorial labelling, as in the Brainbow method[10,11], is not effective in the *C. elegans* embryo, because the speed of development limits temporal accuracy of recombination induction, and the few cell divisions limit opportunities for reporter segregation and differential cell labelling. Photoconversion has been used for single-cell labelling without cell-specific reporters[12,13]; however, under continuous time-lapse imaging, the signal bleaches rapidly, and cytoplasmic labels are poorly suited to visualizing thin neurites.

Here, we present a method for expressing fluorescent reporters or any gene of interest in specific *C. elegans* embryonic neurons, glia or other cell types, without cell-specific drivers. Our method uses an infrared laser to heat a single cell in the embryo and induces gene expression in that cell through heat-shock-response regulatory elements (HREs). This leads to gene expression in small sublineages (one to four cells per embryo).

Previous proof-of-principle implementations of this general strategy[14–19] have not applied this system to track cell or neuronal process growth during embryonic or post-embryonic development. Moreover, previous studies collected only limited biological data. We show that these protocols heat cells to well beyond the physiological range (50–70 °C), or induce heat-independent stress responses, causing extensive damage to cells and preventing neurite outgrowth (see below). Indeed, methods of quantifying cell damage are not described in these studies. By contrast, we develop methodology to perform *in vivo* temperature measurements, and show that our optimized irradiation conditions achieve physiologically compatible temperatures (32–34 °C) with no damage to induced cells or surrounding tissues, which we determine by comparisons to cell-division and neurite-outgrowth timing in non-heated homologous cells.

We performed studies to demonstrate the versatility of our optimized tool for a number of experimental paradigms. We used our setup to label cells and image aspects of NR development. With our system, 5-min irradiation of a single precursor labels progeny cells for 5–6 h, allowing visualization of neuronal and glial cell birth, migration, and neurite outgrowth into the embryonic NR (see Supplementary Fig. 1a for embryonic development timeline). Imaging axon growth dynamics of AVB neurons uncovered a novel retrograde axon-extension mechanism; and imaging amphid neurons in wild-type, *vab-1*/Ephrin and *sax-3*/Robo mutants, revealed a temporal checkpoint for NR axon elongation. Unlike photoconversion or Brainbow, any gene of interest can be driven by HREs, and this strategy can be used to study cell-specific gene function. We investigated the site of action of *daf-6*, a sensory-organ morphogenesis gene, revealing a requirement in glia. Finally, we also used our setup for simultaneous cell ablation and labelling in the same embryo, uncovering glial roles in dendrite extension.

Our setup opens the door to systematic characterization of nervous system assembly in *C. elegans* embryos at subcellular resolution. Since temperature thresholds and dynamics of the heat-shock response are comparable across organisms[20,21], our procedures for calibrating/optimizing infrared-induction parameters should allow immediate extension to other optically/infrared-transparent organisms, including *Drosophila*, zebrafish and *Xenopus*.

## Results

**Infrared-laser microscope setup**. To develop a method for embryonic cell labelling independent of cell-type-specific promoters, we exploited the heat-shock response, which induces transcription of HRE-containing genes following heat exposure. As this response is cell autonomous, heating single cells, using an infrared laser tuned to a 1,455 nm water absorption peak, should allow cell-specific expression of reporters or other genes[16,22] (Fig. 1a). We designed a simple microscope setup for these studies. Laser light is directed to the side port of a microscope (red) and merged with the fluorescent reporter excitation beam (blue), allowing for simultaneous fluorescence imaging and infrared irradiation of cells using a 1.3 NA oil objective (Fig. 1b, Supplementary Fig. 2). A cooling ring connected by tubing to a temperature-controlled water bath controls imaging-objective and sample temperatures.

**Characterizing the heat-shock response in *C. elegans* embryos**. To label cells, we first sought to determine physiological characteristics of the heat-shock response. We generated strains harbouring integrated transgenes expressing either myristoyl-GFP, myristoyl-mCherry or Cre recombinase, under HREs derived from the gene *hsp-16.2*. Expression of the *hsp-16.2* gene locus has been shown to be dependent on heat shock and is strongest in hypodermal cells and neurons[23–25]. The Cre-recombinase transgenics also contain a $his-72_{pro}$::lox-STOP-lox::myr-GFP transgene constitutively expressing myr-GFP once the STOP sequence is excised by Cre-lox recombination[4,26]. The myristoyl moiety directs fluorescent proteins to the cell membrane, enhancing signal in thin neurites. The myr-GFP strain was used to determine the maximal temperature to which embryos can be exposed before damage ensues. Pre-morphogenesis embryos (150–300 min post fertilization) were incubated in a water bath at various temperatures, and hatching rates were used as a proxy for overall cellular health

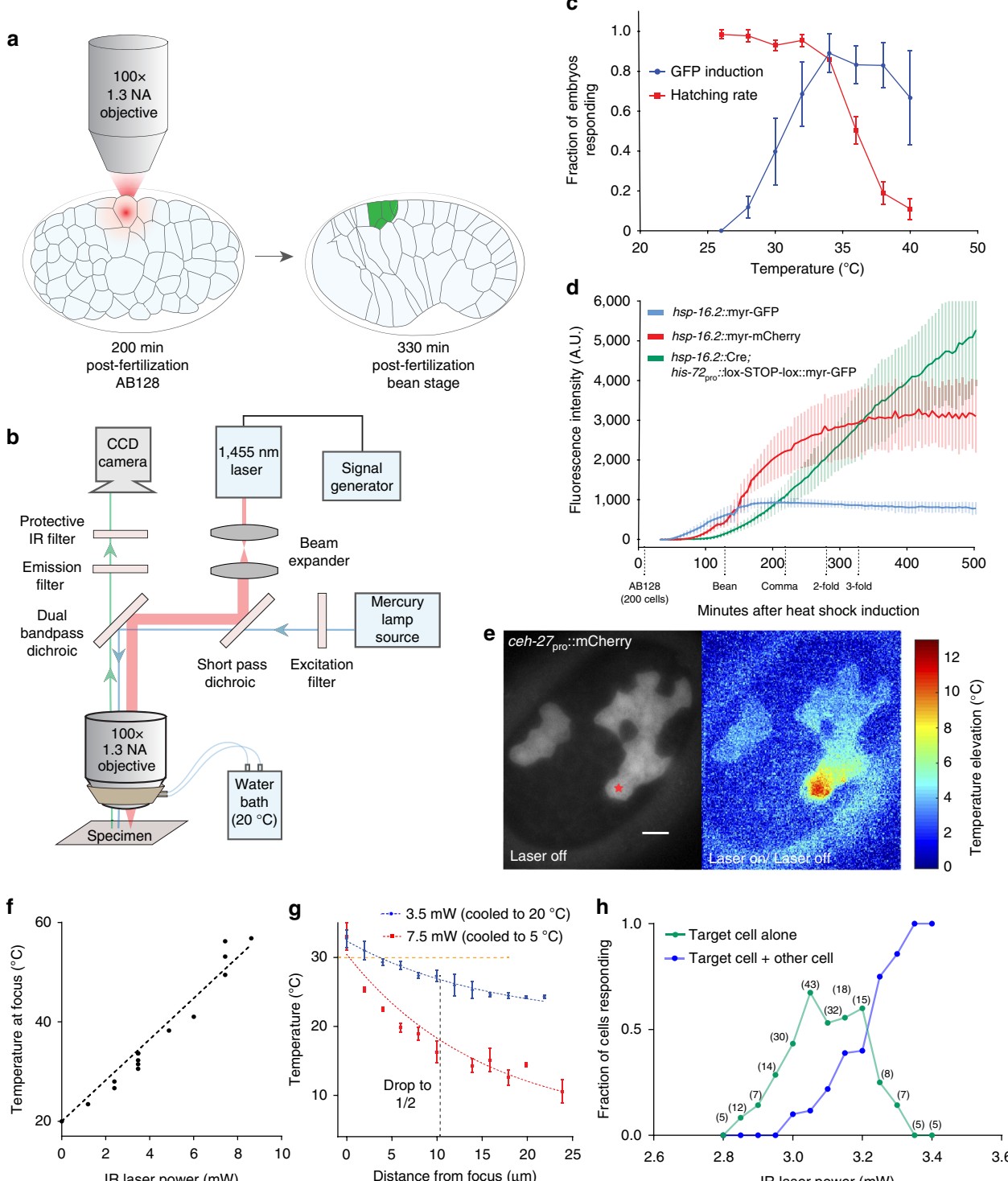

**Figure 1 | Design and temperature calibration of infrared-laser gene induction setup.** (**a**) Infrared-laser induction. Progeny of cells induced at AB128 are visualized by heat-shock-dependent reporters 2 h after induction. (**b**) Infrared microscope design. (**c**) Reporter induction rates and embryonic viability after induction. Induction was scored when >50% of cells showed myr-GFP expression. n>30 for each condition, average of two trials. Error bars, s.d. in response rate across two trials. (**d**) Fluorescent-protein induction kinetics. Average fluorescence intensity after heat induction is plotted versus time (n = 4–6 embryos per reporter). Error bars, standard deviation in fluorescence intensity. (**e**) Fluorescence of *ceh-27*pro::mCherry reporter at ambient temperature at AB128 (left). Thermal map measured in a single embryo with 3.5 mW applied laser power. Star denotes laser focus (right). Scale bar, 5 μm. (**f**) Relationship between temperature at the focus and power applied. Dashed line, linear regression (slope = 4.1 °C mW$^{-1}$). (**g**) Spatial extent of heating in the *x-y* plane. Dashed blue and red lines represent exponential fit of distribution with plateau constrained to the background temperature. Temperature elevation drops to half maximal at ~11 μm from target (black dashed line). Each point represents three to seven temperature measurements taken from individual cells. Orange dashed line, heat-shock threshold. (**h**) Fluorescence induction versus laser power. Irradiation was performed on target cells (ABplaapaa or ABplaapap) for 5 min, and specificity scored using UNC-130-GFP. Number of embryos irradiated for each condition is indicated. IR, infrared.

(more stringent controls were subsequently performed, see below). Animals incubated for 5 min at 32–34 °C produce a robust heat-shock response, and also maintain a high hatching rate (Fig. 1c).

We used these conditions to determine the kinetics of transgene expression in each of the imaging strains we generated. Fluorescence signal is evident in all strains 1–2 h following a 5 min heat pulse in a water bath (Fig. 1d). Signal peak intensity is maintained for 5–6 h for myr-GFP/mCherry transgenics, and until hatching for Cre-lox recombinants. Induction of myr-GFP is fastest, followed by myr-mCherry, and Cre-lox-dependent myr-GFP. GFP imaging using a 488 nm laser results in phototoxicity at imaging intensities 5-fold lower than mCherry imaging[6,27]. Furthermore, mCherry fluorescence reaches a peak intensity 2.5-fold higher than myr-GFP (Fig. 1d). Thus, the maximal myr-mCherry signal without phototoxicity is >12-fold more intense than myr-GFP, and our imaging studies were therefore carried out with myr-mCherry.

**Parameters for single-cell induction using infrared irradiation.** To image neuronal and glial development, we sought to label cells just before morphogenesis, at around 330 min post fertilization at 20 °C (ref. 28). Although infrared-laser induction of the heat-shock response at this stage should yield single-cell labelling, the kinetics of gene expression induction (Fig. 1d) are too slow to observe early neurite/glial-process outgrowth. We therefore irradiated cells at the earlier AB128 cell stage (200-cell embryo), after which cells generate at most four progeny (Fig. 1d, Supplementary Fig. 1a).

To generate a heat distribution raising the temperature of a single AB128 cell to 32–34 °C, while reaching sub-threshold temperatures in neighbouring cells, we first developed a method for measuring the temperature distribution generated by the laser *in vivo*. Previous studies suggest that fluorescent-reporter intensities vary with temperature[16,29,30]. We therefore imaged single cells in embryos carrying a $ceh\text{-}27_{pro}$::mCherry transgene at different temperatures, using a temperature-controlled objective ring (Fig. 1b), and quantified the resulting fluorescence. mCherry signal was linearly dependent on temperature (1.7% decrease in fluorescence per °C between 15 °C and 40 °C) making it a suitable temperature sensor for our study (Supplementary Fig. 3a).

We used embryos carrying the $ceh\text{-}27_{pro}$::mCherry transgene, which is expressed in a thin sheet of cells on the ventral surface of AB128 embryos, as an *in vivo* thermometer (Fig. 1e). This reporter exhibits little out-of-focus signal, allowing accurate measurements in the imaging plane. We found that the temperature rise at the infrared-laser focus depends linearly on laser power (4 °C mW$^{-1}$ over ambient temperature; Fig. 1f, Supplementary Fig. 3b), reaching a steady state within 500 ms and returns to baseline with similar kinetics after turning the laser off (Supplementary Fig. 3c). The temperature distributions around the focus at different laser powers are fit by exponential-decay curves differing only by a multiplicative-scaling factor. Thus, while absolute temperature elevation at all points is proportional to the power applied (Fig. 1f,g), half-maximal heating is always achieved at ~11 µm from the laser focus (Fig. 1g). A similar calibration curve emerges using a nuclear UNC-130-mCherry reporter (Supplementary Fig. 3d). A narrow spatial region can therefore be specifically heated above the heat-shock threshold temperature by lowering the ambient temperature and increasing laser power (Fig. 1g). For *C. elegans* AB128 cells, applying the infrared laser at 3.0 mW for 5 min, while cooling the sample to 20 °C, is predicted, using our calibration curve, to raise the temperature at the laser focus to 32 °C (20 °C + (3.0 mW) × (4 °C mW$^{-1}$)), and this leads to

specific heat-shock reporter expression in 60% of target cells, irrespective of cell location (Fig. 1c,h, Supplementary Fig. 3e). Exceeding this power also results in cell labelling above/below the target, although expression is significantly weaker in non-targeted cells (Supplementary Fig. 3f). Shorter inductions at higher power (2 min, 3.3 W, $n = 24$) or longer inductions at less power (10 min, 2.8 Watts, $n = 18$) do not improve induction frequency. Cells irradiated one generation prior, at AB64, require slightly less power, with higher rates of specificity (85%), probably due to increased cell size (Supplementary Fig. 3g).

Previous studies reported the use of one-second infrared-laser pulses at 11 mW to induce heat shock responses for gene expression in *C. elegans*, zebrafish, medaka and *Arabidopsis*[16,31]. Our calibration experiments suggest that such conditions raise the temperature of the cell at the focus to 65 °C, and therefore likely cause extensive damage. This excessive heating was not reported in these studies, likely because of an error in temperature calibration (Supplementary Fig. 3b). Indeed, when we attempted these conditions in AB128-stage embryos, irradiated cells immediately disintegrated (Supplementary Fig. 3h).

**Cells develop normally after infrared-laser induction.** Heat stress can lead to cellular damage, and the heat-shock response promotes gene-expression changes[32]. However, 24/24 embryos we monitored through embryogenesis hatched, suggesting that global infrared-dependent damage is unlikely. Nonetheless, we aimed to determine whether infrared heating affects irradiated-cell development. We reasoned that tracking axon outgrowth and terminal morphology is a sensitive assay for detecting damage, as many complex events take place to generate a well-formed neuron. We therefore generated a test strain for laser induction harbouring an integrated *hsp-16.2*::myr-mCherry reporter, as well as an $unc\text{-}130_{pro}$::UNC-130-GFP transgene, expressed in 10 easily identifiable nuclei at the AB128 stage, and later in ~40 cells[33]. As a control, we sought to characterize UNC-130-GFP-expressing cells labelled without heat exposure. Although cell-specific reporters are difficult to generate in the embryo, we did find that an $unc\text{-}130_{pro}$::Cre, $dyf\text{-}7_{pro}$::lox-STOP-lox::myr-GFP transgene combination specifically labels the neurons ASI, AWA and ASG, that arise from the UNC-130-GFP-labelled ABplaapap cell. ABplaapap in the test strain was therefore infrared-laser heated, and descendant-neuron development imaged using a spinning-disc-confocal microscope at 1 volume per 5 min. Axon outgrowth in infrared-labelled embryos was assessed and compared with outgrowth in embryos labelled using the $unc\text{-}130_{pro}$::Cre, $dyf\text{-}7_{pro}$::lox-STOP-lox::myr-GFP transgene combination (Fig. 2a). As shown in Fig. 2b, no differences in axon-outgrowth timing, growth rate or length at the 2-fold stage were seen (see Supplementary Fig. 4 for stage determination). We also introduced an $odr\text{-}10_{pro}$::GFP reporter, expressed in differentiated AWA neurons, into the UNC-130-GFP strain. Post-embryonic expression of $odr\text{-}10_{pro}$::GFP as well as neuron morphology of AWA neurons whose precursor was irradiated were indistinguishable from the non-heat-shocked bilateral counterpart (9/9 animals; Fig. 2c).

In these studies, we observed a slight delay (27% increase in cell cycle time, on average) in division timing of heat-exposed cells compared with their bilateral counterparts. For example, while parental cells of the left and right AWA neuron divide, on average, within 3 min of each other in untreated embryos, embryos in which the left AWA grandparental cell was heat-exposed exhibited an average delay of 17 min in the terminal division compared with non-heated right homologues (Fig. 2d). In cells irradiated one generation prior, a 23 min average delay was noted. Nonetheless, in all the labelled embryos, axon

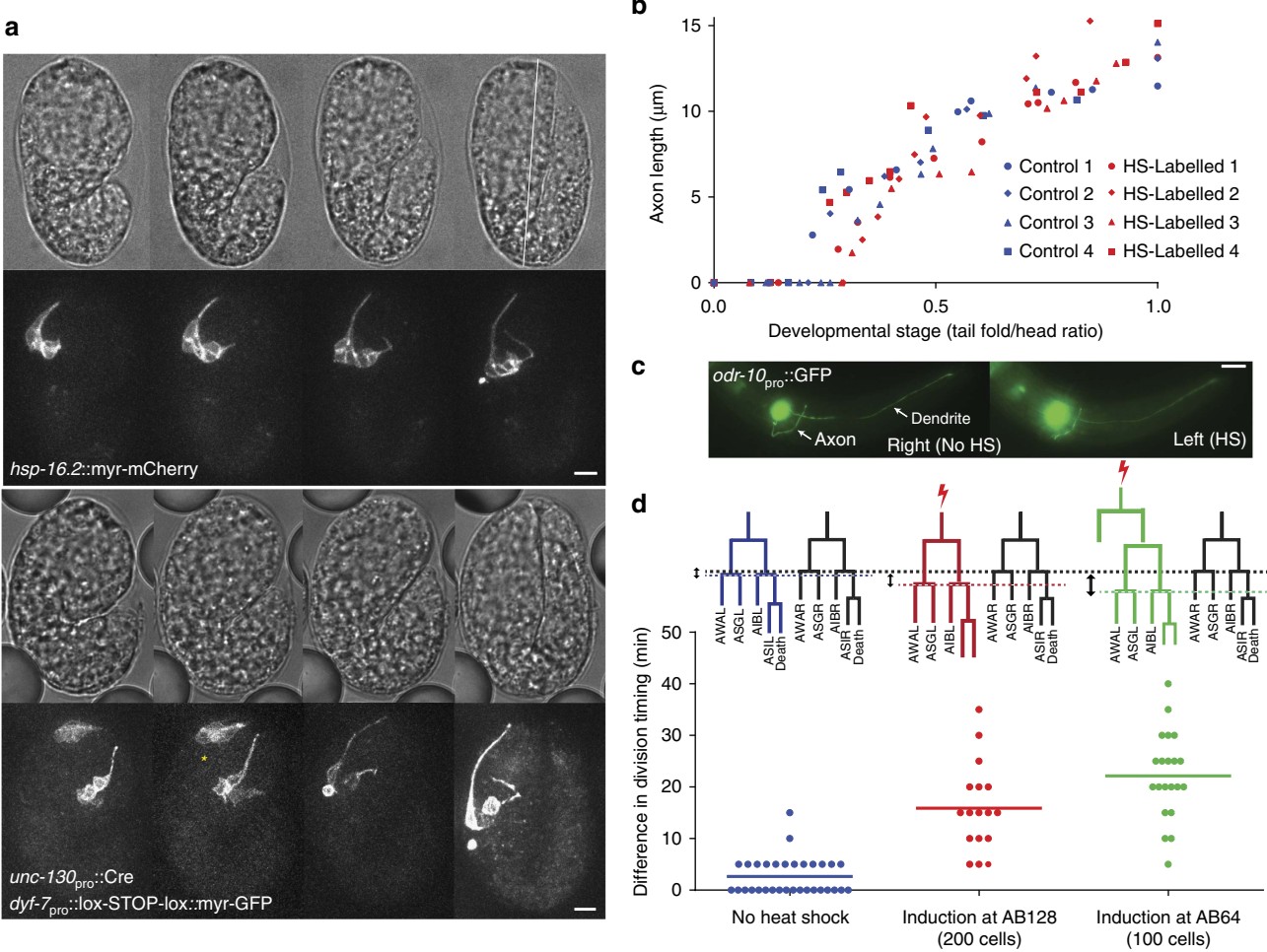

**Figure 2 | Development of heat-shock-labelled cells is normal. (a)** Time-lapse imaging of infrared-laser labelled and promoter-labelled cells for axon-outgrowth comparisons. Asterisk, nonspecific expression from recombinase activity. Scale bar, 5 μm. **(b)** Timing of axon initiation and growth is not affected by heat-shock induction. Axon length plotted against developmental stage in infrared-laser-labelled ASI, AWA, ASG, AIB neurons and promoter-labelled ASI, AWA and ASG neurons. $n = 4$ embryos per reporter. **(c)** Post-embryonic fate marker expression and axon trajectory in the AWA neuron are not affected by heat shock (HS). Induction performed at AB64 in ABplaapa. Animals scored at L3/L4 stages. Scale bar, 10 μm. **(d)** Cell-division timing is modestly delayed in heat-shocked cells. Timing of final division of heat-shocked cells compared with contralateral homologues using the symmetric UNC-130-GFP marker. Measurements in the 'no heat shock' condition are absolute differences between left and right sides. $n > 8$ embryos for each.

outgrowth timing was not affected. Furthermore, the natural variation in cell-division timing between bilateral cells, which can exceed 10 min, also does not lead to axon-outgrowth timing differences in descendant neurons (Supplementary Fig. 5).

Together, our data suggest that even though heat treatment can result in a modest delay in cell division, this has no effect on onset or progression of axon outgrowth. This validates the use of our setup to study embryonic cell morphogenesis, and reveals that neuron generation and differentiation timings are independently controlled.

**Specific labelling of neurons and glia by infrared gene induction.** We used the *hsp-16.2*::myr-mCherry; *unc-130*pro::UNC-130-GFP strain to determine whether different cell types can be labelled using our setup. As the *C. elegans* lineage is invariant, descendants of any precursor are known, and their identities can be verified by their terminal morphology, division and migration patterns[27] (Fig. 3b,c). Irradiating different precursor cells at the AB128 stage successfully labels progeny cells of diverse types, and following labelled cells using time-lapse microscopy allows

visualization of their development (Fig. 3). For example, the sensory neurons ASG, AWA and ASI are labelled by irradiating the ABplaapap precursor, as is the interneuron AIB (Fig. 3d). Likewise, the URB sensory neuron, the ILso and AMsh glial cells, and the hyp3 epithelial cell are labelled following infrared laser heating of the ABplaapaa precursor (Fig. 3e). Other neurons, glia and epithelial cells are labelled by irradiating other precursor cells (Fig. 3f–h).

Previous studies demonstrated that sense-organ neurons are born near the nose tip and extend anchored dendrites by posterior migration of the cell soma[3,13]. Time-lapse imaging of labelled cells confirmed this retrograde extension of amphid sensory dendrites ($n = 12$; Fig. 3d, Supplementary Movie 1) and revealed that the same mechanism operates for AMsh ($n = 10$) and AMso ($n = 4$) amphid glia, and for neurons and glia associated with other sensory organs (URB and CEM neurons, ILso glia; Fig. 3e,f).

Time-lapse imaging of sparsely labelled cells allowed tracking of other developmental events. For example, the CEM neuron dies in hermaphrodites and survives in males. We found that this neuron extends a dendrite before cell death

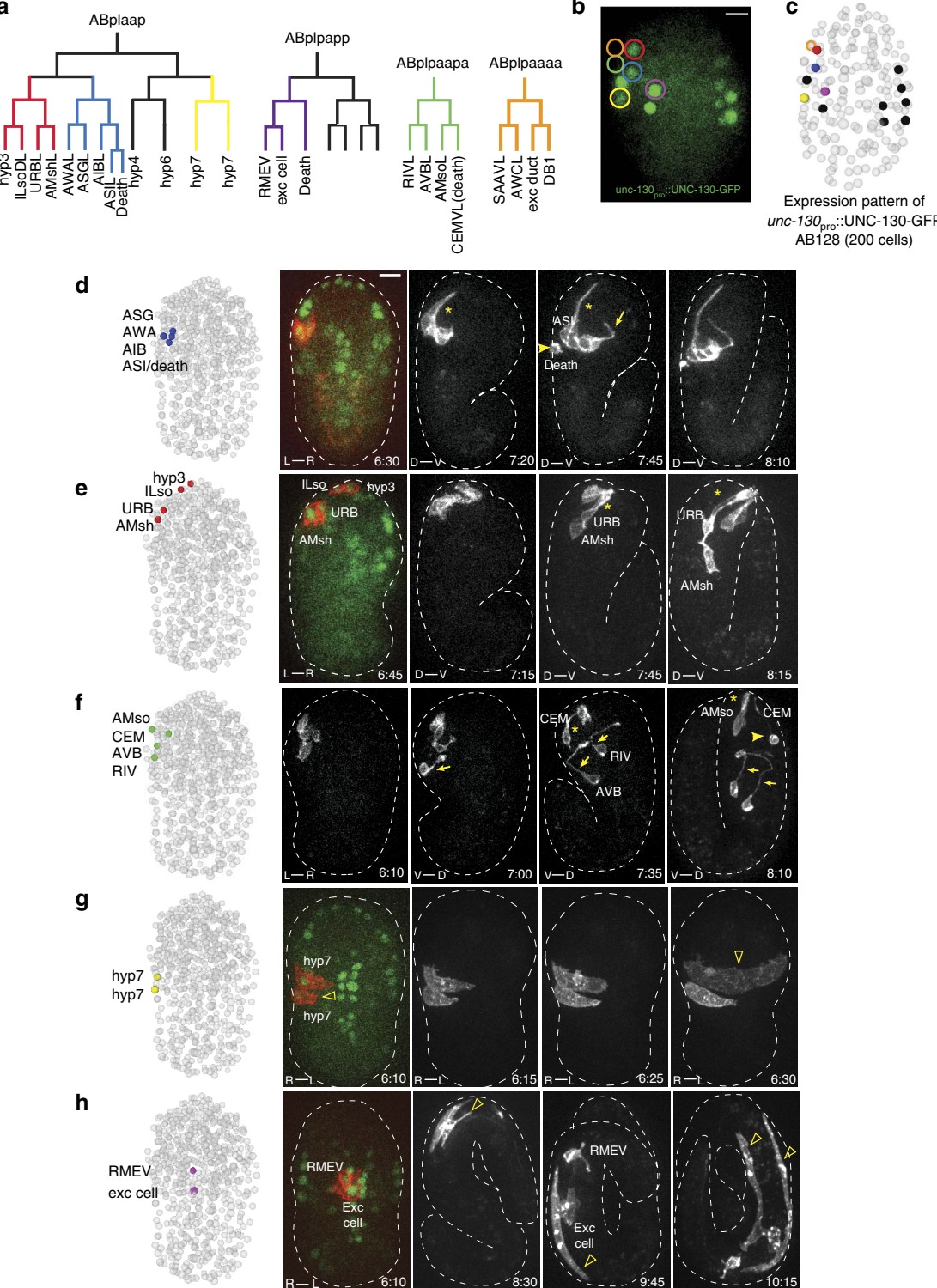

**Figure 3 | Infrared-laser-induction allows visualization of morphogenesis of diverse cell types.** (**a**) Cell lineages in which induction was performed. Colours correspond to experiments shown in **d**–**h** and Supplementary Movies 1–3. ABplpaaaa induction (orange) is shown in Supplementary Movie 3. (**b**) Expression pattern of UNC-130-GFP nuclear marker at AB128. Circled cells were targeted for labelling in separate experiments (**d**–**h**). Scale bar, 5 µm. (**c**) UNC-130-marked cells are shown on coordinates from a representative fully lineaged embryo[27] at AB128. (**d**) Amphid neurons undergo retrograde dendrite extension (star) and axon extension ventrally and then dorsally into the nerve ring (arrow). ASI sister-cell death is visible (arrowhead). See Supplementary Movie 1. (**e**) Sensory glia and neurons are born near the nose-tip and extend retrograde processes (star). (**f**) Proximal axon segment of AVB and RIV (arrow), CEM-neuron early stages of death (arrowhead) are visible. See Supplementary Movie 2. (**g**) Hypodermal cell migration/fusion (open arrowhead). (**h**) Excretory cell elongation (open arrowhead). The (**d**–**h**) Labelled cells are shown on coordinates from a fully lineaged embryo after the final division is complete[27] (left). The embryo axis of the imaging plane is shown at the bottom left of images. (D–V, dorsal–ventral; R–L, right–left). Note that the embryo turns 90° in the eggshell during morphogenesis (Supplementary Fig. 1). Times are given relative to comma stage (07:10; ref. 3). Scale bar, 5 µm.

ensues (Fig. 3f, Supplementary Movie 2), an observation not previously reported. Time-lapse imaging of single hyp7 hypodermal cells revealed that they extend filopodia stretching towards the ventral midline, eventually fusing with an unlabelled hyp7 cell approaching from the opposite side ($n = 5$; Fig. 3g). myr-mCherry rapidly diffuses to label the entire binucleate cell (Fig. 3g). We also followed elongation of the excretory cell (Fig. 3h), which takes place well into the 3-fold stage, demonstrating the perdurance of our reporter ($n = 5$).

These results show that our setup can label different cell types for a prolonged time to uncover dynamic aspects of embryonic cell development.

**Single cell imaging reveals a novel axon-extension mechanism.** Targeting the infrared laser to the ABplpaapa precursor, which neighbours UNC-130-GFP-expressing cells, results in subsequent labelling of two neurons, AVB and RIV, whose development had not been characterized. We therefore followed their growth using time-lapse imaging to determine how their final morphologies emerge (Fig. 3f, Supplementary Movie 2, $n = 4$). At 06:40 post fertilization at 20 °C, the AVB neuron projects a short dorsal axon. Further dorsal growth into the NR then ceases for 30 min. At the same time, the AVB neuron cell body pivots around the stationary axon, and migrates posteriorly and dorsally away from the NR, extending the cell-body proximal portion of the axon. While this retrograde axon extension takes place, a large growth cone, comparable in size to the cell body, is evident at the axon tip, perhaps tethering the axon to resist the migrating cell-body pull (Supplementary Movie 2). The distal portion of the axon then resumes dorsal growth to complete its trajectory by 07:50 post fertilization. Remarkably, this mode of axon growth is also used by the co-labelled RIV neuron. Over a period of 35 min (06:45–07:20 post fertilization), RIV extends an axon dorsally into the NR. Once growth is complete, the cell body migrates posteriorly, laying down the proximal axon segment.

These observations demonstrate the ability of our setup to uncover new aspects of embryo development, and demonstrate that axons, and not only dendrites, can grow by retrograde extension. A similar mode of axon growth has been described during cerebellar granule cell development in vertebrates. Here, axons bifurcate, extend processes in the densely packed molecular layer, and their cell bodies then migrate radially, laying down proximal axon segments[34]. Thus, AVB and RIV neurons may be suitable for studying this poorly understood guidance process.

Of note, AVB was previously proposed to pioneer NR formation[35], as were RME neurons[36]. However, our single-cell imaging shows that the RIV (Fig. 3f; Supplementary Movie 2; NR growth at 06:55) and SAAV neurons (Supplementary Movie 3; NR growth at 06:30) enter the NR before AVB (Supplementary Movie 2; NR growth at 07:25) or RME axon growth (Supplementary Fig. 6; NR growth at 07:30), suggesting that these latter cells do not initiate NR formation.

**Single-axon growth dynamics in axon-guidance mutants.** Many genes controlling axon guidance in *C. elegans* are known, and most are conserved in vertebrates[37–41]. Nonetheless, how neuronal-guidance defects in mutants of these genes arise is generally unknown, as mutants are usually examined postembryonically. Moreover, guidance mutants often exhibit embryonic lethality; thus, previously unappreciated defects may exist in embryos that do not survive. We therefore sought to use our infrared-labelling system to characterize single-axon guidance defects in developing embryos. We chose to investigate mutants in the *sax-3*/Robo and *vab-1*/EphR genes and to examine axon-navigation dynamics of four amphid sensory neurons,

AWA, ASI, ASG and AWC, in which postembryonic defects were previously described[37,40].

We first characterized the wild-type axon-outgrowth pattern of the left AWA, ASI and ASG neurons by irradiating their common grandparental precursor (Fig. 3d; Supplementary Movie 1). After dendrite extension, all three neurons extend axons ventrally in the early 1.5-fold stage (07:30), then turn dorsally to enter the NR, with full axon extension observed by 08:15 ($n = 8$). Irradiating the ABplpaaaa precursor cell labels the left AWC neuron, as well as the SAAVL and DB1 neurons, and the excretory duct cell. The AWC axon follows a growth sequence similar to AWA/ASI/ASG (Supplementary Movie 3), but initiates ventral growth much earlier (06:45; $n = 4$).

The *sax-3*/Robo mutants exhibit several postembryonic defects in amphid sensory-neuron trajectories[37] (Table 1), however, defect a etiologies are not known. For example, *sax-3* animals have anterior axons that do not incorporate into the NR. These axons could grow at a normal rate, but fail to interact with the NR; or may initially fail to grow, and only extend processes much later, as has been shown for other mutants[42,43]. We found that in all the *sax-3* mutant embryos with anterior processes, the timing of axon initiation is unaffected, suggesting that *sax-3* acts specifically to regulate axon incorporation into the NR (Fig. 4a,b). Similarly, other postembryonic defects in these mutants appear to arise by failed navigation, rather than axon-outgrowth timing defects (Fig. 4b). In some embryos with aberrant axon trajectories, axons initially branch and later retract one of the branches (4/11). These dynamics were not previously described, and suggest the existence of key decision nodes along the axon migration path (Fig. 4a).

The *sax-3* mutants also have an anteriorly displaced NR, but unlike other guidance problems, this defect does not appear to have an early origin. Indeed, anterior NRs are observed in 15% of *sax-3* larvae (Table 1), but are not evident in early stages of NR formation (Fig. 4c). Thus, in *sax-3* mutants, the NR forms in the correct location relative to the embryo as a whole, but is later displaced, perhaps during embryo elongation.

Besides the previously catalogued defects in *sax-3* mutants, we observed previously undescribed abnormalities in dendrite placement. In one embryo, dendrites initially extended along the left–right axis before correcting their trajectory to extend along the anterior–posterior axis (Supplementary Fig. 7a). Another embryo exhibited defasciculation of amphid neuron dendrites (Supplementary Fig. 7b).

Unlike *sax-3* mutants, *vab-1* mutants have a single guidance defect in which amphid neuron axons fail to project ventrally, instead entering the NR laterally[37] (Table 1). We found that these abnormal axons still terminate axon growth at the dorsal part of the NR, as in wild-type animals; however, axon length is shorter (Fig. 4d,f; $n = 4$). Remarkably, unlike wild-type axons, laterally projecting axons do not immediately turn dorsally when contacting the NR. Rather, a delay in dorsal turning is evident (Fig. 4e,f). This suggests that the NR has a competence period for the navigation of some axons. It is possible, for example, that axons require guide processes to grow, and if these have not formed, axon growth will stall.

Together, these studies demonstrate the utility of our setup for quantitative comparisons of axon guidance between embryos and between genetic backgrounds, and unmask previously unappreciated checkpoints governing nervous-system assembly.

**AMsh glia-specific rescue of *daf-6* mutants.** Infrared-laser heating can be used to express any gene of interest in a specified cell at a specified time, and can therefore be used to investigate

**Table 1 | Embryonic and postembryonic imaging for *sax-3* and *vab-1* mutants.**

| | Phenotype | *sax-3(ky123)* AWA postembryonic *n > 50* | *sax-3(ky123)* embryonic *n = 31* | *vab-1(dx31)*, AWC postembryonic, *n > 50* | *vab-1(dx31)* embryonic *n = 27* |
|---|---|---|---|---|---|
| | Wildtype | 43% (23) | 52% (12) | 63% (35) | 76% (16) |
| | Lateral Axon | 13% (7) | 26% (6) | 38% (21) | 24% (5) |
| | Anterior Axon | 22% (12) | 22% (5) | 0% (0) | 0% (0) |
| | Axon Termination | 6% (3) | 0% (0) | 0% (0) | 0% (0) |
| | Anterior Nerve Ring | 15% (8) | 0% (0) | 0% (0) | 0% (0) |
| | Arrest Prior to NR Entry | | (6) | | (6) |
| | Dendrite Misplacement/ Breakage | | (2) | | |

Embryonic defects were characterized by labelling the AWA, ASI and ASG precursor at AB128 or AB64 (ABp[l/r]aapap or ABp[l/r]aapa). Postembryonic defects were characterized in L3/L4 animals using an AWA-specific marker for *sax-3(ky123)*, *odr-10* $_{pro}$::GFP and an AWC-specific marker, *odr-1* $_{pro}$::YFP, for *vab-1(dx31)*.

gene function. We used our setup to determine the site of action of the *daf-6* gene, required for sensory-organ morphogenesis. In *daf-6* mutants, amphid sensory organs are enlarged, and sensory neurons project into extracellular vacuoles surrounded by AMsh glia, and are not exposed to the environment[44] (Fig. 5a). This precludes neurons from taking up dye from the surrounding medium, providing a convenient assay for sensory-organ integrity. *daf-6* is expressed in AMsh and AMso glia and functions in the embryo during sense-organ formation[45]. To determine whether *daf-6* function in AMsh glia is sufficient, we irradiated the grandparental precursor of the right AMsh glial cell (Fig. 5b) in animals carrying the UNC-130-GFP reporter and a *hsp-16.2*::*daf-6* cDNA transgene. Eight out of 27 operated animals were rescued for the dye-filling defects on the heat-treated side of the animal, while neurons on the non-heated side failed to fill with dye (Fig. 5c,d). This demonstrates that *daf-6* activity in AMsh glia is sufficient for sensory-organ morphogenesis.

**Combined ablation/imaging shows glia control dendrite shape.** *C. elegans* cell function is often studied by laser ablation. Typically, 440 nm pulsed dye lasers are used to kill cells, perhaps through DNA damage[46,47]. The effects of cell ablation on the development of a neighbouring cell can be used to determine whether adjacent cells signal to each other.

We explored the use of our infrared laser to ablate cells, by exposing them to excessive heat, and to uncover interactions between adjacent cells. Previous studies, as well as experiments presented here, demonstrate that *C. elegans* amphid sensory

neurons grow dendrites by retrograde extension (Fig. 3d). These studies raise the possibility that AMsh glia, which tightly associate with these dendrites, promote dendrite anchoring as cell-body migration ensues. However, because early embryonic promoters specific for the AMsh glia are not known, this idea has not been directly tested.

To test this hypothesis, we modified our infrared-irradiation parameters to ablate the grandparental cell of the AMsh glial cell in animals carrying the UNC-130-GFP and *hsp-16.2*::myr-mCherry reporters. Simply increasing laser power led to off-target reporter-gene induction. Recent studies have shown that pulsing the laser at high frequency reduces heat accumulation away from the focus, allowing for targeting specificity[17,48]. We found that pulsing the laser at 6 Hz for 10 s, with a pulse duration of 8.3 ms at 13 mW, prevents off-target induction (Fig. 6a). Furthermore, targeted cells fail to divide to the final round by 200 min after ablation (12/12 embryos). myr-mCherry reporter expression is sometimes detected in the ablation-targeted cells using power levels of 10–12 mW, but increased power prevents this, likely because the cell dies immediately. Importantly, the timing of the final cell division in neighbouring amphid neuron precursors is only modestly affected (14 min average delay; Fig. 6b), suggesting that our conditions specifically damage the target cell without affecting surrounding cells.

In six ablated embryos, we induced myr-mCherry expression in amphid neuron precursors using our standard conditions (Fig. 6c). In 3/6 embryos, associated amphid neurons initially formed a dendritic protrusion, but this was not anchored and was dragged posteriorly as the cell migrated (Fig. 6d). This

observation reveals that AMsh glia or glia-secreted proteins are likely to contribute to dendrite anchoring. As three cells in addition to the AMsh glia cell are also eliminated in our experiments, we cannot rule out that these cells (URB neuron, IL sheath glia, hypodermis) may contribute to the anchor, although none of these cells are in contact with the amphid dendrite in the post-embryonic structure.

These studies highlight the extended applications of our setup, and demonstrate its use for tracking single cells in animals that have been functionally manipulated.

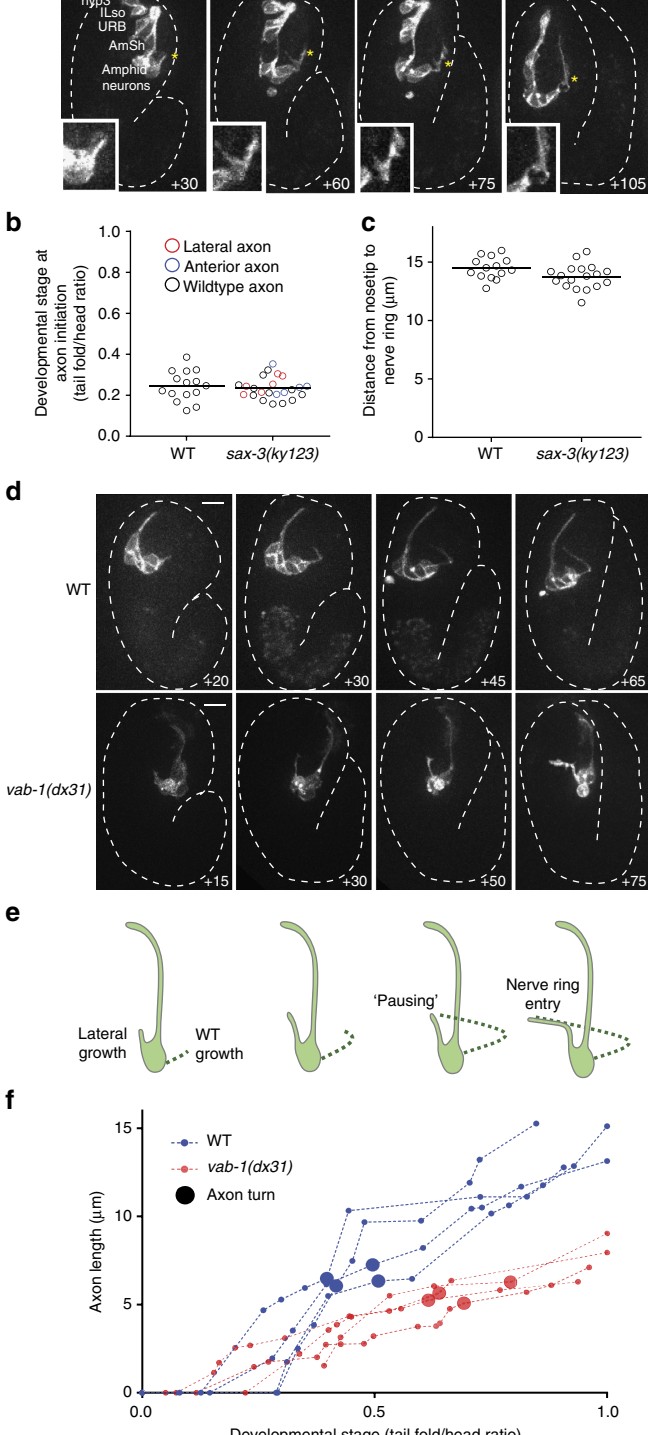

## Discussion

We present a method for visualizing neural-circuit development and manipulating cell function in vivo in C. elegans embryos. Our method is versatile and can be used to label neurites for imaging, to drive gene expression, and to ablate cells. We demonstrate applications of this setup and reveal biological phenomena that were not previously reported. For example, we observe a novel retrograde mechanism of axon outgrowth in the AVB and RIV interneurons; we provide evidence that a timing/competence mechanism controls axon-outgrowth dynamics in the NR; we demonstrate that the DAF-6 protein functions in AMsh glia to regulate sensory-organ formation; and we show that AMsh glia are required for dendrite anchoring during retrograde extension.

We envision that our setup could be integrated with existing lineaging software[4,49] to automatically label or manipulate cells with minimal manual input. Indeed, recent improvements in computer algorithms now allow real-time nuclear tracking in C. elegans embryos[50,51]. Thus a complete description of C. elegans nervous system assembly on a cell-by-cell basis should be feasible. We envision the following pipeline: (i) real-time lineage tracking of animals expressing a pan-nuclear GFP reporter would be performed on embryos from the 4- to 200-cell stage. (ii) Infrared-laser labelling or perturbation would then be automatically induced in cells of interest by feeding the appropriate nuclear coordinates to the infrared-laser alignment system. (iii) The cells could be visualized over time using mCherry fluorescence time-lapse imaging at the start of morphogenesis and using GFP to track nuclear positions of other cells in the embryo.

The images and movies presented here were acquired using a spinning-disc confocal microscope, for which rapid volumetric time-series acquisition is not possible. For example, although we were able to see axon forking and retraction in sax-3 and vab-1 mutants, we could not follow the behaviour of single branches given our 5-min time resolution. Also, our measurements of axon outgrowth were less reliable after the onset of twitching due to motion-induced blurring. However, image quality of infrared-labelled embryos could be improved using other imaging methods, such as selective-plane-illumination microscopy, which is suited for rapid live imaging with reduced phototoxicity. Using selective-plane-illumination microscopy, embryo volumes can be acquired at 30 × the speed of confocal microscopy, allowing blur-free imaging even in twitching embryos[8].

A few limitations of our system should be considered in the context of using conventional, promoter-based reporters when available. One caveat is the reliance on the heat-shock

**Figure 4 | Infrared-laser induction allows characterization of embryonic guidance defects in sax-3 and vab-1 mutants.** (a) Embryonic imaging of anterior amphid axon growth and failure to turn into the nerve ring in sax-3(ky123). Induction performed at AB64 in ABplaapa. Inset: magnified image of starred region showing forking and retraction of axons during growth. Time, minutes from comma stage. (b) The initiation time of axon outgrowth is not affected in sax-3 mutants. n = 15 embryos for WT, n = 23 embryos for sax-3(ky123). (c) Anterior nerve-ring placement is not pronounced in sax-3 mutants. Distance measured from nose tip to nerve ring in 2-fold embryos. n = 15 embryos for WT, n = 18 embryos for sax-3(ky123). (d) Embryonic imaging of amphid neurons in WT for comparison and vab-1(dx31) showing lateral trajectory. Time as in a. (e) Model for timing compensation in nerve-ring entry in vab-1 mutants. (f) Axon trajectories during nerve-ring entry are shorter in vab-1 mutants, and the dorsal turn occurs later. n = 4 embryos each for wild-type and vab-1(dx31).

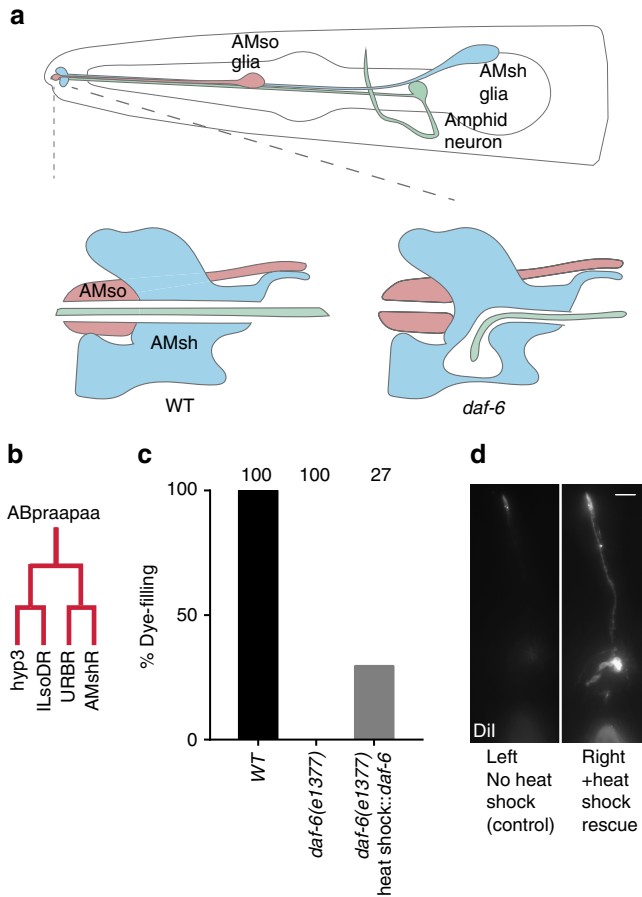

**Figure 5 | Infrared-laser induction of *daf-6* in AMsh glia rescues amphid channel development.** (**a**) Amphid channel structure in wild-type and *daf-6(e1377)* mutants. (**b**) AMsh-glia lineage starting at AB128. (**c**) Rescue of dye-filling defects by infrared-laser induction. Scale bar, 10 μm. (**d**) Dye-filling defects rescued on the right side of the animal (heat-induced) and not the left side (no heat induction).

response, which may alter cell physiology. While we observed modest delays in division timing of heat-shocked precursors, these differences had no effect on subsequent morphogenesis in the neuron we studied. However, previous work has suggested that pathways that buffer against heat stress may act differentially across neuron types and may affect neuronal migration, polarization and axon outgrowth[52,53]. Further improvements to lower the heat-shock-response threshold, by cultivation at colder temperatures or overexpression of heat-shock factor, may mitigate any cell division delay. Another limitation is that gene-expression onset following irradiation of some terminally differentiated *C. elegans* cells may be too slow to visualize their early development. We solve the issue here by labelling precursor cells, but this may not offer sufficient resolution for some applications. Improvements in the design of fluorescent proteins to allow faster folding could alleviate this limitation. Finally, the procedure for embryo isolation and irradiation requires additional time for each embryo (~3 h) before morphogenesis, limiting use for large-scale studies.

Although there are many advantages to sparse cell imaging, an important remaining challenge is to compare spatial/temporal data acquired for different cells in different embryos to establish relationships among these cells. In our comparisons, we align embryos temporally based on overall embryo morphology to compare axon growth rate, but we do not compare spatial

dynamics of neurites or cell bodies between embryos. Recently, a method using the lattice of seam cells as fiducial markers has been used to untwist the embryo during elongation and twitching, allowing comparison of CAN, ALA and AIY neurons between different embryos[54]. Adding these alignment markers to our labelling strains and using the established pipeline could allow us to perform these comparisons. Moreover, our imaging of axon growth suggests that much of NR development occurs before twitching (Fig. 3); thus, simpler methods using rigid registration with sparse markers may be sufficient for generating composite data sets (Peter Insley and S.S., in preparation).

In summary, we describe a simple inexpensive method for cell labelling, gene-function and cell-function studies without the need for cell-specific drivers. Given that heat-shock thresholds are similar across organisms, we expect that our induction conditions and strategy for optimization could be adapted readily to other transparent embryos including *Xenopus*, *Drosophila* and zebrafish.

## Methods

**Strains and plasmids.** Animals were cultured on NGM agar seeded with OP50 bacteria at 20 °C using standard methods[55]. Transgenes/plasmids/mutants are listed in Supplementary Tables 1-3.

**Embryo mounting and recovery.** For infrared labelling and ablation experiments, embryos were dissected from hermaphrodites and transferred by mouth pipette onto glass slides with M9 and 20-μm polystyrene beads as slide spacers, as previously described[56]. This places embryos under slight compression and causes stereotypical turns of embryos in the eggshell during morphogenesis (Supplementary Fig. 1). The slides were sealed to prevent evaporation using vaseline heated to 40 °C, which cools quickly on the slide and does not induce the heat shock response. For scoring *daf-6* gene rescue and verifying AWA morphology postembryonically, embryos were induced on 2% agarose pads to facilitate recovery. Approximately 2 h after heat-shock induction, the embryos were transferred to agar plates with a wire pick. The embryos were allowed to hatch on plates and grow until L3 or L4 before scoring.

**Temperature calibration for embryo heat shock and viability.** Embryos were dissected from hermaphrodites and allowed to develop for ~2 h in M9. Embryos were then transferred into PCR tubes and heat shocked for 5 min using a temperature gradient program on a Thermocycler with 2 °C intervals. Embryos were subsequently recovered and either mounted on agar pads for scoring heat shock induction by myr-GFP, or placed on plates and scored for hatching after 24 h.

**Kinetics of fluorescent protein induction.** The embryos were mounted on slides using 20 μm bead spacers, as above. The slides were then put in a thin-bottomed plastic container and placed in a water bath at 33 °C for 5 min. After heat shock, the slides were imaged using spinning-disk confocal microscopy at 5 min intervals (see Imaging Procedures), and the time from the heat shock was recorded. The total fluorescence was quantified in a single slice over time for four to six embryos per transgene and then averaged in every time point to generate the kinetics curves. As the images were taken at different power levels depending on the laser line, the resulting fluorescence intensities were normalized by the applied laser power to create a comparison, which should be affected only by protein levels and quantum efficiency between the different fluorescent proteins.

**Optical design and alignment for infrared irradiation experiments.** A 1,455 nm continuous wave infrared laser (RLR-2-1455, IPG Photonics Corp.) was coupled to a wide-field fluorescent microscope (Zeiss AxioImager A1). After 4 × beam expansion, the beam was directed into a side port and a shortpass dichroic (T860spxrxt_1500, Chroma Tech.) was used to merge the epifluorescent light path, first filtered with a dual excitation filter (59022 ×, Chroma Tech.) and the infrared laser beam. A dual dichroic (59022bs, Chroma Tech.) along with single-wavelength emission filters (GFP, XF3080; mCherry, XF3081 Omega Optical) was used for imaging GFP/mCherry simultaneously with infrared induction. An additional shortpass filter (FESH1000, ThorLabs) was used to protect the camera and eyepiece from small amounts of infrared irradiation. Infrared induction experiments and temperature measurements were performed using a Zeiss oil immersion objective, 1.3 NA (1,018-595). A signal generator (BK Precision, 4054) was connected to the external interface of the laser to control the duration and frequency of pulses for ablation and for synchronizing the camera frame acquisition with the laser induction for temperature measurements. Fluorescence images for temperature measurement were acquired using a

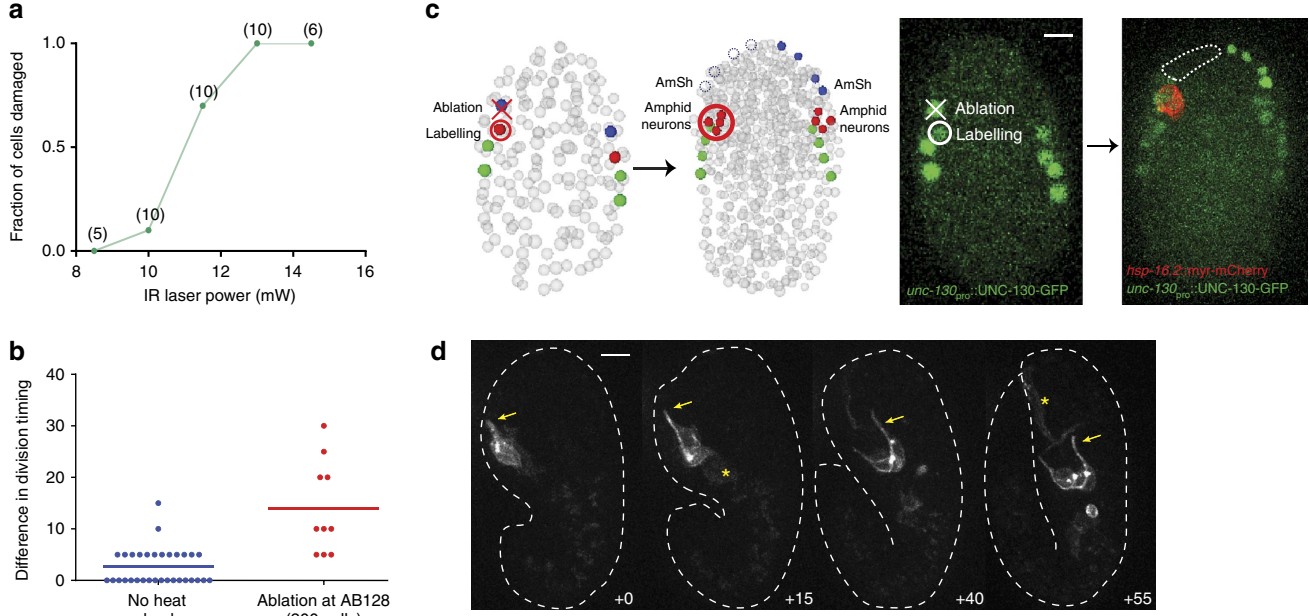

**Figure 6 | Combined infrared ablation/labelling shows glia control dendrite extension.** (**a**) Damage induced versus laser power with infrared ablation parameters. Number of animals shown on top. (**b**) Delay in division timing in amphid-neuron precursors at the final division round of divisions from infrared ablation of the AMsh glia precursor (ABplaapaa). $n = 5$ embryos. (**c**) Left, cell ablation and labelling strategy. Right, representative image showing absence of AMsh glia and sister cells after ablation, and labelling of amphid neurons (red). Scale bar, 5 μm. (**d**) Time-lapse imaging of amphid neurons after AMsh glia ablation. Dendrite begins to extend but fails to anchor. Asterisk, nonspecific labelling from heat shock. Scale bar, 5 μm. IR, infrared.

CoolSNAP HQ2 CCD camera (Photometrics). A temperable ring (Pecon, 0269-010) was fastened to the objective and connected to a circulating water bath (VWR scientific) with heating and cooling control, for sample temperature control. Pictures of the optical setup can be found in Supplementary Fig. 2.

To identify the focus of the laser, we found that the infrared laser could be detected at high exposure times (∼1 s) on the CCD camera, and this was used for x-y alignment. The z-alignment was determined using optical trapping of 3 μm polystyrene beads (Polysciences, Inc).

**Sample cooling and mCherry temperature-dependence analysis.** For background cooling during induction experiments and for mCherry temperature calibration curves, the temperature of the sample was controlled by heating and cooling the immersion objective, which is in contact with the specimen. To confirm that the sample temperature matched the water bath, a thermocouple (Physitemp) was inserted directly into the sample and compared. For water bath temperatures ranging from 15 °C to 40 °C, the sample was within <1.5 °C of the water bath temperature.

For calibrating mCherry fluorescence, the water bath temperature was changed between 15 °C and 40 °C and images were captured of a single cell expressing mCherry in anaesthetized larval animals (ceh-27pro::mCherry, expressed only in four to six neurons postembryonically). We noted that the z-focus changed during temperature changes due to expansion/contraction of the objective, and appropriate compensations were made to keep the cell in focus. The total fluorescence of the cell was measured after background subtraction by summing a region of interest circumscribing the entire cell in ImageJ, and the fluorescence was plotted at different temperatures, normalized to the 20 °C value. Some temperatures measurements were revisited for each sample after temperature alterations to confirm reversibility of the fluorescence change.

**Temperature measurements in embryos.** The laser was focused at the centre of an AB128 mCherry-expressing cell, and fluorescence images were acquired one second after the laser was turned on and one second after the laser was turned off. To generate temperature maps, the two images were down-sampled by 3× to reduce noise, and background subtracted. The processed images were divided (laser ON image/laser OFF image), and ratiometric fluorescence decrease was converted to temperature increase based on the measured mCherry temperature sensitivity. Heat map was rendered using MATLAB.

To measure temperature elevation at the focus and in neighbouring cells, total cell fluorescence (Supplementary Fig. 2d) was quantified in the laser ON/OFF images after background subtraction using ImageJ, and the resulting values converted to temperature elevation as above. To plot spatial distribution, distances between the centre of the cell of interest and the focus of the laser were measured. Data from multiple embryos ($n = 3$–5 per power setting per

reporter) was aggregated, and independent temperature measurements from cells within 2 μm were averaged. Error bars represent standard deviation.

**Time-lapse imaging procedures for embryos.** Time-lapse images were acquired on the CellVoyager 1000 (Yogokawa/Olympus) spinning-disk confocal microscope, using a 1.35 NA ×100 Olympus silicone oil-immersion objective (UPLSAPO 100XS). Z-stacks of 25–30 slices at 0.8–1 μm spacing were acquired every 5 min with 200 ms exposures per slice. Laser-power levels passing through the objective were measured at 20 μW on the 488 laser (GFP) and 100 μW on the 561 laser (mCherry). After twitching, the exposure time was decreased to 40 ms to decrease motion blur, and mCherry laser power was increased to 500 μW. Temperature varied between 21 °C and 24 °C during acquisition, a major cause of developmental timing variation during morphogenesis.

**Induction and scoring of embryos for labelling and ablation.** The embryos were mounted using 20 μm bead spacers or 2% agarose pads. During heat-shock induction, the cell was irradiated for five continuous minutes. To determine power versus response curve, the precursor cells were identified using a nuclear UNC-130-GFP marker. A marked cell (ABpl/raapaa or ABpl/raapap at AB128, ABpl/raapa at AB64) was induced for 5 min in different embryos at various powers. After 150 min, heat-shock-driven mCherry induction in progeny cells was scored. If four cells were labelled and the signal co-localized with UNC-130-GFP, induction was scored as specific. If additional cells (usually also four additional cells, not co-localizing with GFP) were labelled, embryos were scored as having off-target induction.

For ablation experiments, the AMsh-glia precursor was identified (ABpl/raapaa) by UNC-130-GFP at AB128. The cell was irradiated and scored as ablated if it failed to divide to the final round 200 min after induction.

**Axon-outgrowth measurement.** Neurite length was quantified in ImageJ by marking a set of points along the axon and summing the Euclidean distance between these. The distance between neighbouring points $(X,Y,Z) \rightarrow (X', Y', Z')$ was calculated by $D_{3D} = [(D^2_{X-X'} + D^2_{Y-Y'} + D^2_{Z-Z'})]^{1/2}$. This distance was plotted as a function of developmental stage (tail-to-head length; Supplementary Fig. 3) to facilitate comparison between embryos. The tail-to-head ratio ranges from 0 (comma stage) to 1 (2-fold stage). Note that after twitching begins (tail-to-head = 0.5–0.6), some time points were omitted if the axon was not visible.

**Dye-filling assays.** The embryos were recovered after laser irradiation, grown until the L4 stage, and placed singly in drops of M9 + 5 μg ml⁻¹ DiI for 30 min (ref. 44). The animals were recovered onto plates and imaged with wide-field microscopy.

**3D renderings of embryogenesis.** Nuclear coordinates were taken from a fully lineaged embryo[27] at 202 min (AB128) and 357 min (early morphogenesis), and 3D models with coloured sublineages were rendered in MATLAB.

**Data availability.** The data that support the findings of this study are available from the corresponding author on request.

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

## Acknowledgements

We thank Shaham Lab members for discussions and comments on the manuscript. We thank Peter Insley, Wolfgang Keil, Alexander Katsov, Anthony Santella, Pavak Shah, Varun Narendra and Rockefeller University Bio-Imaging for technical support. A.S. was supported by NIH Medical Scientist Training Program grant T32GM07739 to the

Weill Cornell/Rockefeller/Sloan-Kettering Tri-Institutional MD-PhD Program. S.S. was supported by NIH grants HD078703, NS064273 and NS081490.

## Author contributions

A.S. and S.S. developed the method, designed the experiments and wrote the manuscript. A.S. performed the experiments and analyses.

## Additional information

**Competing financial interests:** The authors declare no competing financial interests.

