## [Peer Review File · Nature Communications]

Reviewers' comments:

Reviewer #1 (Remarks to the Author):

In this paper Singhval & Shaham describe the systematic use of cell-specific, laser induced gene expression using a heat inducible promoter. The authors explain and standardize their method, focusing on embryonic stages of the nematode *C. elegans*. In addition, the authors demonstrate the usefulness of their approach by describing previously unknown aspects of neuronal circuit assembly in early embryos of both wild type and mutant embryos of *C. elegans*. Although laser-induced activation of transcripts driven by heat-shock promoters is not new per se and has been described and used in the literature, the authors go a step further and illustrate how to adapt this technique to visualize early developmental events that could be difficult or otherwise impossible to follow by conventional cell reporters. To this end, the authors use elegant lineage tracing approaches (with CreLox mediated excision of a stop cassette) to permanently or transiently label small subsets of the lineage with fluorescent tracers. These approaches allow spectacular high resolution imaging of neuro-developmental processes in live animals, like we have not seen before. The authors further show that their new methodology can be combined with time-tested laser ablation approaches, further expanding the toolbox of visualization and manipulation. The experiments are rigorously controlled and demonstrate that their heat shock approach has only limited effects on the cells adjacent to the heat shocked cells. Overall, the manuscript represents an important technical improvement that will surely be followed by more detailed explanations of the mechanisms underlying processes such as axon or dendrite outgrowth, attachment or retrograde extension. Strengths of the manuscript include the high resolution imaging of early neurodevelopmental events as well as the rigorous control of all experiments and parameters. Since embryonic imaging of neurodevelopmental processes has for a number of reasons been very challenging in *C. elegans* this work represents a significant advance. It would have been nice (although maybe not required) to see a little bit more about what could have been done with this technique, findings that go beyond the relatively limited (and somewhat disconnected) findings made by the authors.

Major points to be addressed:

1. Careful analysis shown by the authors demonstrates that use of "physiologically-compatible temperatures" is a key factor to succeed in activation of IR-induced transgenes without producing collateral damage or off target labeling/expression. Nonetheless, the authors detect slowing down of the cell cycle, which they argue has no effect on otherwise normal development. However potential situations where cells are susceptible to changes within this range of "compatible" temperatures need to be discussed such as the existence of pathways that buffer axon guidance errors against mild thermal stress (Wang et al., *Neuron*. 2013 79(5):903-16.). Thus, the use of traditional reporters whenever available may be advisable to rule out unknown cell-specific responses to temperature changes.
2. Authors must clearly explain the limitations inherent to their approach. The approach involves the rather time-consuming dissection of hermaphrodites to obtain embryos, and moderate to low success rates (60% of neurons respond, line 172 page8) for heat-shock induction. Further, combination of labeling and ablation perturbations impact negatively the number of experimental observations than can be obtained in a given period of time (I assume that this is reflected in the relative low number of animals analyzed (the number of animals analyzed is actually quite low)). Thus, a careful, cost benefit analysis must be considered before engaging in this type of experimentation.
3. The authors claim that AMsh glia are responsible for dendrite anchoring of amphid neurons. The ablation experiment targets the grandparental cell of the AMsh glia cell, which means according to fig

5b indicates that 3 other cells are killed. A potential role of these ablated cells in amphid dendrite anchoring is not addressed and cannot formally be ruled out.

4. The paper represents an extension of an existing method (Kamei et al. 2009 6, 79 Nature Methods). The visualization of early occurring events during development is an interesting improvement that should allow researchers to address different biological questions previously considered too technically challenging. The biological observations are a good complement of what can be analyzed at these specific early stages. However, they are in essence more descriptive than explanatory. While some new insights into various aspects of neurodevelopment were gleaned from the described experiments, the method is still in search of a major question.

5. The statement on p.4 lines 74-76 that cell-specific heat shock experiments have not been used "to study any aspect of nervous system development, even post-embryonically", and that "...none of these previous studies was used to collect novel biological data..." is false. Harris et al., Development (1996) used cell specific, heat shock-induced (by a laser beam) expression of the transcription factor *mab-5* in Q cell descendants (the precursors to a set of neurons) to establish the non-autonomous function of this transcription factor in neuronal cell migration, i.e. neural development.

Minor points:

Page6 line 119: Include a rationale why *hsp-16.2* was selected as the heat-shock promoter.

Page8 line 158: Supplementary Fig. 1e does not exist. The authors probably refer to Fig. 1e

Figure 1c and 1d. What do the bars represent in these figures?

Figure 4a and 4d. control images of non-defective animals are needed for comparisons.

Reviewer #2 (Remarks to the Author):

In this paper, the authors reported new tracking method for development of single cell in *C. elegans* using IR laser-induced heat shock response mechanism. The manuscript is well written and the data are almost sufficiently analyzed. I therefore suggest its publication in Nature Communications. However, the authors should consider the following several minor suggestions:

1. It was noticed that, the present work looks like a natural extension of a series of former works carried out before by other groups, although not completely the same, especially in terms of their applying of IR laser-induced HSP system. This would be reduced the originality of the paper. The authors should clearly describe the advantages of the prepared laser system in introduction part in comparison with other previous researches.

2. We consider that lower temperature (32-34°C) for gene expression is one of the key points as a breakthrough for solving some biological secrets in the future. However, there is nothing information why they can dramatically reduce the temperature although they used the same gene code as the previous reported. I am wondering why such a low temperature can make the expression.

3. The HSP-mediated gene expression requires quite time to generate accurate proteins. Actually, the authors incubated long time at least for 150 min for confirming protein inductions. Certain evidences and more comments to improve the problem should be clearly provided.

4. The authors irradiated for only 5min to express the proteins with the fixed laser power. Is it optimal

time and power? Do they have more detail information for the optimal condition?

5. The authors just demonstrated cell-division to sure the biocompatibility of the laser irradiation. More serious evaluations on the safety concern of cell damages are necessary.

6. For the biomedical applications of HSP-promoted gene regulation, a series of pioneering efforts have been made before. To name a few, the following literature closely related to the current work can be cited for the readers to better understand the status in the emerging area.

Miyako, E. et al. Photothermic regulation of gene expression triggered by laser-induced carbon nanohorns. PNAS 109, 7523 (2012).

Reviewer #3 (Remarks to the Author):

Singhal & Shaham present a method for controlled heating of *C. elegans* samples with cellular resolution that is considerably better characterized than previous efforts. They establish the usefulness of their calibrated technique to induce gene expression with minimal side effects, and furthermore make several fascinating and novel observations about cell dynamics during development. The technique should interest the wide readership of Nature Communications. The paper and figures are beautifully put together, and I have no suggestions for improvement, however minor.

Reviewer #1 comments:

Strengths of the manuscript include the high resolution imaging of early neurodevelopmental events as well as the rigorous control of all experiments and parameters. Since embryonic imaging of neurodevelopmental processes has for a number of reasons been very challenging in C. elegans this work represents a significant advance. It would have been nice (although maybe not required) to see a little bit more about what could have been done with this technique, findings that go beyond the relatively limited (and somewhat disconnected) findings made by the authors.

We are happy that the reviewer appreciates the rigorous validation of our tool and found the imaging to be striking. As described in our comments to the editor, the primary aim of this manuscript is to demonstrate the broad utility of our method to study *C. elegans* embryogenesis at a cellular level, which has been largely inaccessible due to the general lack of conventional fluorescent reporters to follow single cells during this period. While we expect that more detailed characterization of mechanisms will now be possible with this system, we believe that additional studies are beyond the scope of this paper, as we have already rigorously validated this tool and demonstrated its use in uncovering a number of new cellular behaviors.

1. Careful analysis shown by the authors demonstrates that use of “physiologically-compatible temperatures” is a key factor to succeed in activation of IR-induced transgenes without producing collateral damage or off target labeling/expression. Nonetheless, the authors detect slowing down of the cell cycle, which they argue has no effect on otherwise normal development. However potential situations where cells are susceptible to changes within this range of “compatible” temperatures need to be discussed such as the existence of pathways that buffer axon guidance errors against mild thermal stress (Wang et al., Neuron. 2013 79(5):903-16.). Thus, the use of traditional reporters whenever available may be advisable to rule out unknown cell-specific responses to temperature changes.

We appreciate this very important point raised by Reviewer #1. In our manuscript, we have assessed with great care the potential cell damage in irradiated cells, using a number of strategies to look for defects. These include:

(1) comparison of the timing and rate of axon outgrowth (Fig. 2a,b) in heat-shock labeled embryos vs. embryos labeled with a combinatorial reporter

(2) maintenance of expression of a post-embryonic sensory receptor (Fig. 2c)

(3) the final morphology of the axon and dendrite (Fig. 2c).

(4) We also demonstrate that natural variability in the timing of the cell cycle (~10 minutes) between left-right homologous cells appears to have no effect on the timing of axon outgrowth (Supplementary Fig. 5e).

Nonetheless, as recommended by the reviewer, we have now included a section in the Discussion (page 20 of the revised manuscript) to address this point, and have also suggested improvements to design of the heat shock genetic constructs that could mitigate the division delay.

2. Authors must clearly explain the limitations inherent to their approach. The approach involves the rather time-consuming dissection of hermaphrodites to obtain embryos, and moderate to low success rates (60% of neurons respond, line 172 page8) for heat-shock induction. Further, combination of labeling and ablation perturbations impact negatively the number of experimental observations than can be obtained in a given period of time (I assume that this is reflected in the relative low number of animals analyzed, the number of animals analyzed is actually quite low). Thus, a careful, cost benefit analysis must be considered before engaging in this type of experimentation.

We agree that IR laser-labeling can require time investment for gathering information on large sets of embryos under some circumstances. However, we make three observations:

(1) Even in studies using conventional reporters (e.g. Wu et al., 2011; Christiensen et al., 2015), <6 embryos were analyzed in experiments requiring long-term time-lapse imaging that included quantification of cellular behavior. Thus, it is not our method *per se* that is the issue, but the long-term imaging. Indeed, when full imaging was not required, we routinely looked at >60 embryos for each experiment (Table 1, Fig. 1h, Supp. Fig. 1e,g)

(2) Because the manipulations in our method are cell specific, this reduces substantially the number of animals that need to be observed to achieve significance of the results.

(3) As we have noted in the Discussion, although we currently perform the process manually, we envision that future integration of the IR tool with computer-automated cell identification methods will allow for labeling and imaging of embryos with minimal user input.

At the reviewer's suggestion, we now address these issues in the Discussion (page 20 of the revised manuscript).

3. The authors claim that AMsh glia are responsible for dendrite anchoring of amphid neurons. The ablation experiment targets the grandparental cell of the AMsh glia cell, which means according to fig 5b indicates that 3 other cells are killed. A potential role of these ablated cells in amphid dendrite anchoring is not addressed and cannot formally be ruled out.

We thank the Reviewer for bringing up this concern. As noted, three other cells are killed in our ablation experiments—the URB neuron, the IL sheath glial cell, and the hypodermis cell 3 (hyp3). Both the URB neuron and IL sheath cells are contained in other sensilla and not in physical contact with the amphid dendrite channel, we find that a role for these cells in anchoring the amphid dendrite is highly unlikely. A role for the hyp3 cell in providing an additional supporting structure through contact with these glial cells is formally plausible, although the cell is not in direct contact with the neurons we observe.

At the suggestion of the reviewer, we have now revised the text to address this possibility (page 18 of the revised manuscript).

4. The paper represents an extension of an existing method (Kamei et al. 2009 6, 79 Nature Methods). The visualization of early occurring events during development is an interesting improvement that should allow researchers to address different biological questions previously considered too technically challenging. The biological observations are a good complement of what can be analyzed at these specific early stages. However, they are in essence more descriptive than explanatory. While some new insights into various aspects of neurodevelopment were gleaned from the described experiments, the method is still in search of a major question.

Please see our response to the editor, and our first response to Reviewer 1.

5. The statement on p.4 lines 74-76 that cell-specific heat shock experiments have not been used “to study any aspect of nervous system development, even post-embryonically”, and that “...none of these previous studies was used to collect novel biological data...” is false. Harris et al., Development (1996) used cell specific, heat shock-induced (by a laser beam) expression of the transcription factor mab-5 in Q cell descendants (the precursors to a set of neurons) to establish the non-autonomous function of this transcription factor in neuronal cell migration, i.e. neural development.

We appreciate the Reviewer’s point that laser-induced gene expression has been used previously in a study for establishing the role of *mab-5* in neuronal migration and have revised the text accordingly (page 4 of the revised manuscript) and now include this reference.

However, it is important to note that there are a number of concerning features of this experiment, which is why we chose not to reference this study in our initial manuscript. The authors of this study note that the laser intensity producing the desired rescue sometimes caused blocking of the subsequent cell division of the Q neuroblasts and sometimes prevented the cell from migrating entirely in the posterior Q neuroblast lineage (QR.p), an effect independent of the heat-shock driven transgene. Although the result demonstrating cell-autonomy of *mab-5* still holds, it is clear that the method as described could not be used for visualizing wild-type

development. Moreover, the conditions are not useful for producing gene induction without damage in all sublineages, given that the QR.p cells exhibit aberrant behaviors after irradiation. Our improvements appear to surmount the issues of cell cycle arrest or migration defects in the embryonic cells that we have studied.

Minor points:

Page6 line 119: Include a rationale why hsp-16.2 was selected as the heat-shock promoter.

We have included a brief explanation in the text (page 6 of the revised manuscript).

Page8 line 158: Supplementary Fig. 1e does not exist. The authors probably refer to Fig. 1e

We thank Reviewer 1 for identifying this error, which has now been corrected.

Figure 1c and 1d. What do the bars represent in these figures?

In Figure 1c, the bars represent the standard deviation of the fraction of embryos responding (>30 embryos for each trial, with 2 trials for each condition). In Figure 1d, the bars represent the standard deviation in the fluorescence intensity across the embryos quantified (n=4-6 embryos for each reporter).

We have added information to the figure legends to explain the error bars.

Figure 4a and 4d. control images of non-defective animals are needed for comparisons.

We have added a panel of wild-type images to Figure 4d to facilitate comparison between developmental stages with *vab-1* mutants. For Figure 4a, the only relevant time point of comparison would come from the initial stage of axon outgrowth, since the final anterior process placement does not depend on a time-lapse series. Therefore, we have chosen to associate the wildtype images with Figure 4d, rather than 4a.

Reviewer #2 comments:

1. It was noticed that, the present work looks like a natural extension of a series of former works carried out before by other groups, although not completely the same, especially in terms of their applying of IR laser-induced HSP system. This would be reduced the originality of the paper. The authors should clearly describe the advantages of the prepared laser system in introduction part in comparison with other researches.

While the concept of targeted IR laser gene induction is not new, we have found that practical and theoretical errors in the previous characterization of this system have prevented its broad application. We have made several substantive improvements that now, for the first time, make the method practical:

(1) We demonstrate that the application of previously published induction parameters cause extensive cell damage in the embryo (Supplementary Fig. 3h), and heat cells to temperatures far exceeding physiological heat shock temperatures (~65°C, Supplementary Fig. 3b).

(2) We develop a strategy for measuring temperatures *in vivo* in *C. elegans* (Fig. 1e) and re-optimize the IR-laser system to reach physiologically relevant heat shock thresholds without cellular damage (Fig. 1e-g).

(3) Previous studies have not established rigorous methods to quantify damage to cells. We devise such methods to identify developmental defects.

We have now revised the Introduction (page 4 of the revised paper) to make our achievements more clear.

2. We consider that lower temperature (32-34°C) for gene expression is one of the key points as a breakthrough for solving some biological secrets in the future. However, there is nothing information why they can dramatically reduce the temperature although they used the same gene code as the previous reported. I am wondering why such a low temperature can make the expression.

We thank the Reviewer for bringing up this important point. Previous studies have suggested that the heat shock response is dependent on the duration of heat stress and the temperature, with higher temperatures required when short periods of heat stress are used (Gerner 1987; Tulapurkar et al., 2009). Previous IR laser implementations irradiated cells for very short periods of time, and as we show in Supplementary Fig. 3b, the laser powers used in these studies raises the temperature of cells to ~65°C. We also show that the kinetics of the heating process are such that the heat is maintained at the focus only for the duration of the irradiation, such that after the laser is turned off, the temperature drops back to baseline rapidly (Supplementary Fig. 3b). Thus, an irradiation time of 1 second, as used in Kamei et al. 2009, means that the cell is only heated for 1 second in total. Though these conditions may elicit the heat shock response in some cell types (as shown in Kamei et al. 2009), we find that these conditions frequently damage cells, causing them to disintegrate immediately (Supplementary Fig. 3h).

A crucial advance in our work is to heat cells over much longer time periods (5 minutes) at much lower, physiologically-relevant (32-34°C) heat shock temperatures. This duration/temperature combination provides the same “dose” of heating while preventing the damage that results from heating cells to very high temperatures.

We have now included a brief additional explanation to the text associated with Supplementary Fig. 3b.

3. The HSP-mediated gene expression requires quite time to generate accurate proteins. Actually, the authors incubated long time at least for 150 min for confirming protein inductions. Certain evidences and more comments to improve the problem should be clearly provided.

As the reviewer points out, visualization of fluorescent protein labels following irradiation of cells requires time. Actual times are shorter than those quoted by the reviewer: ~100 minutes

using myr-mCherry reporters (Fig. 1d), and ~60 min for GFP (Fig. 1d). This is not a problem with our system *per se*, but with fluorescent reporter folding. Future improvements in the design of fluorescent reporters, to allow faster folding and brighter fluorescence, could alleviate this limitation.

We have added comments in the Discussion (page 20 of the revised manuscript) to suggest how this time delay can be improved.

4. The authors irradiated for only 5min to express the proteins with the fixed laser power. Is it optimal time and power? Do they have more detail information for the optimal condition?

We have observed that irradiation protocols within narrow ranges are roughly equivalent (2-10 minutes of heating, reaching 30-34°C), with lower temperature irradiations requiring longer irradiation times and vice versa, as suggested by the idea of the “thermal dose” in Point 2. For the experiments shown, we focused on the use of one irradiation condition (5 min, ~3 mW), and performed all control experiments (Figure 2) and subsequent induction experiments with this condition to minimize variability.

We have added a comment (page 9 of the revised manuscript) to address this.

5. The authors just demonstrated cell-division to sure the biocompatibility of the laser irradiation. More serious evaluations on the safety concern of cell damages are necessary.

We disagree with this comment. We actually used a number of additional strategies to look for defects in irradiated cells, and all strategies demonstrated no visible defects:

- (1) comparison of the timing and rate of axon outgrowth (Fig. 2a,b) in heat-shock labeled embryos vs. embryos labeled with a combinatorial reporter
- (2) maintenance of expression of a post-embryonic sensory receptor for the neuron AWA (Fig. 2c)
- (3) the final morphology of the axon and dendrite (Fig. 2c).

Nonetheless, we agree that more discussion could be provided and have done so in the Discussion (page 20 of the revised manuscript).

6. For the biomedical applications of HSP-promoted gene regulation, a series of pioneering efforts have been made before. To name a few, the following literature closely related to the current work can be cited for the readers to better understand the status in the emerging area.

Miyako, E. et al. Photothermic regulation of gene expression triggered by laser-induced carbon nanohorns. PNAS 109, 7523 (2012).

We appreciate the Reviewer’s comment, and find the work cited to be very relevant, and have now cited the reference.

Reviewer #3 comments:

Singhal & Shaham present a method for controlled heating of C. elegans samples with cellular

resolution that is considerably better characterized than previous efforts. They establish the usefulness of their calibrated technique to induce gene expression with minimal side effects, and furthermore make several fascinating and novel observations about cell dynamics during development. The technique should interest the wide readership of Nature Communications. The paper and figures are beautifully put together, and I have no suggestions for improvement, however minor.

We thank Reviewer 3 for the positive feedback on our manuscript.

REVIEWERS' COMMENTS:

Reviewer #1 (Remarks to the Author):

The authors have addressed the major points raised by this reviewer. I appreciate that their studies do indeed focus rather on breadth of possible applications than depths to investigate a particular question in more detail. I can accept this as a reasonable explanation. I believe that the strengths of the paper, including rigor, technical advance, breadth of application, and (relative) ease outweigh the lack of depth. I further agree with the authors that an in depth application for a single question is beyond the scope of this paper and would not make much sense. However, maybe a paragraph, in analogy to their response to the editor, could be added after validation of the method to clarify for the reader that what follows is not merely a series of assorted experiments, but the systematic application of the method to answer different types of biological questions during early development; maybe adjust abstract and intro accordingly. I think that this would underscore the utility of the method and better show the breadth of questions that can be answered with their method.

Minor points:

p.4, line 79: "temperature" instead of "temperatures"

Reviewer #2 (Remarks to the Author):

Shaham et al. perfectly responded for my comments. I'm fully satisfied with the current version. Thus, I'd like to recommend that the paper should be accepted.

Response to reviewers:

Reviewer 1.

However, maybe a paragraph, in analogy to their response to the editor, could be added after validation of the method to clarify for the reader that what follows is not merely a series of assorted experiments, but the systematic application of the method to answer different types of biological questions during early development; maybe adjust abstract and intro accordingly. I think that this would underscore the utility of the method and better show the breadth of questions that can be answered with their method.

We have addressed this point by restructuring the introduction to explain that our experiments are designed to demonstrate the versatility of the method for studying embryogenesis (page 5 of revised manuscript).